# Synthesis of Phenylboronic Acid-Functionalized Magnetic Nanoparticles for Sensitive Soil Enzyme Assays

**DOI:** 10.3390/molecules27206883

**Published:** 2022-10-14

**Authors:** Can Li, Zhishang Shi, Jinxing Cai, Ping Wang, Fang Wang, Meiting Ju, Jinpeng Liu, Qilin Yu

**Affiliations:** 1National & Local Joint Engineering Research Center on Biomass Resource Utilization, College of Environmental Science and Engineering, Nankai University, Tianjin 300350, China; 2Key Laboratory of Molecular Microbiology and Technology, Ministry of Education, Department of Microbiology, College of Life Sciences, Nankai University, Tianjin 300071, China; 3Institute of Agricultural Resources and Environment, Ningxia Academy of Agro-Forestry Science, Yinchuan 750002, China

**Keywords:** magnetic nanoparticle, soil enzyme, protein capture, magnetic enrichment

## Abstract

Soil enzymes, such as invertase, urease, acidic phosphatase and catalase, play critical roles in soil biochemical reactions and are involved in soil fertility. However, it remains a great challenge to efficiently concentrate soil enzymes and sensitively assess enzyme activity. In this study, we synthesized phenylboronic acid-functionalized magnetic nanoparticles to rapidly capture soil enzymes for sensitive soil enzyme assays. The iron oxide magnetic nanoparticles (MNPs) were firstly prepared by the co-precipitation method and then functionalized by (3-aminopropyl)triethoxysilane, polyethyleneimine and phenylboric acid in turn, obtaining the final nanoparticles (MNPPBA). Protein-capturing assays showed that the functionalized MNPs had a much higher protein-capturing capacity than the naked MNPs (56% versus 6%). Moreover, MNPPBA almost thoroughly captured the tested enzymes, i.e., urease, invertase, and alkaline phosphatase, from enzyme solutions. Based on MNPPBA, a soil enzyme assay method was developed by integration of enzyme capture, magnetic separation and trace enzyme analysis. The method was successfully applied in determining trace enzyme activity in rhizosphere soil. This study provides a strategy to sensitively determine soil enzyme activity for mechanistic investigation of soil fertility and plant–microbiome interaction.

## 1. Introduction

As a complex organic whole, soil, especially agricultural soil, is of great significance to human social development. Soil enzymes, such as urease, acid phosphatase, invertase, dehydrogenase and catalase, are important components in the soil ecosystem and are widely involved in the material cycle and energy metabolism of soil. Soil enzyme activities are affected by soil properties, tillage methods, environmental factors and other factors, especially total nitrogen, organic matter and available phosphorus, etc. For example, long-term fertilization can significantly increase the contents of soil total nitrogen and active organic nitrogen, as well as the activities of soil urease, alkaline phosphatase and urease, but fertilization may inhibit catalase activity [1,2,3].

The changes of these enzyme activities are of great importance to reflect the growth status and interaction mechanism of soil microorganisms and plants and reveal their response mechanism to environmental factors [4,5]. Most of the time, these interactions occur around the roots of plants, along with the changes of soil enzymes. However, the detection of trace substances is often difficult [6]. The same challenges also appear in the preservation, pretreatment and detection of trace soil enzymes [7]. These limitations affect the measurement of trace soil enzymes to different degrees. The detection methods of soil enzyme activities are mature. However, when dealing with trace enzymes, the effectiveness of the detection method encounters difficulties [8,9,10]. Improving the concentration of enzymes in the sample by optimizing the pretreatment method to meet the requirements of the detection method has become an effective technical path.

Magnetic enrichment technology has shown promising application prospects in many fields such as biochemistry, environmental chemistry, food chemistry and physical chemistry. First, materials synthesized using magnetic nanoparticles can be rapidly enriched in an applied magnetic field but can be dispersed without settling in a non-magnetic environment [11,12]. Secondly, the surface of magnetic nanoparticles can be loaded with functional groups through different synthesis methods, such as amino and carboxyl groups and other epoxy groups [13,14], which can be used for protein capture, separation and other processes [15,16]. For example, Alinezhad et al. [17] used magnetic nanoparticles combined with MSPE technology to separate, enrich and detect three kinds of NSAIDs in environmental water samples. The operation was simple, and the analysis time was short, with a recovery rate of 93.6–98.9%. Liu et al. [18] prepared a novel *N*,*N*-dimethyldodecamine-functionalized magnetic enrichment material (Fe_3_O_4_@MDHM) and used it for the enrichment of flavonoids in grape juice. However, for different enrichment objects, the structure, composition and properties of magnetic materials also have a great impact on the enrichment and analysis results [19,20].

Therefore, this paper hopes to synthesize functional magnetic nanoparticles to efficiently capture soil enzymes and construct a rapid recovery of nanoparticles and trace soil enzyme activities detection method, as shown in Figure 1. Then, a set of efficient and accurate soil enzyme activities is formed for a better study of the soil ecosystem.

## 2. Results

### 2.1. Synthesis of the Phenylboronic Acid-Grafted Magnetic Nanoparticles

The magnetic nanoparticles were first prepared by the co-precipitation method [21] and characterized by TEM observation and XRD analysis. As shown in Figure 2a, the initial MNPs had round-like morphology, with sizes of 10–15 nm. XRD analysis further revealed that the MNPs had the standard XRD spectrum of the Fe_3_O_4_ crystal (Appendix A), indicating pure Fe_3_O_4_ nanoparticles. After grafting PEI and PBA [22,23], the obtained nanoparticles, i.e., MNPPEI and MNPPBA, exhibited round-like morphology and sizes similar to the initial MNPs (Figure 2a). Zeta potential analysis further showed that the MNPPEI nanoparticles had higher Zeta potential than the initial MNPs (45 mV versus 24 mV, Figure 2b), which was attributed to the strong positive charges of PEI on the surface of MNPPEI. Moreover, MNPPBA exhibited lower Zeta potential than MNPPEI (Figure 2b), indicating the grafting of PBA partially reduced the positive charges of PEI [24]. FT-IR analysis confirmed that the presence of C-NH-C and CH groups in MNPPEI and the presence of the B-O group in MNPPBA (Figure 2c).

### 2.2. Efficient Protein Capture by the Synthesized Magnetic Nanoparticles

Owing to the presence of phenylboronic acid groups, the MNPPBA nanoparticles may have a high protein-capturing capacity. To verify this, the model protein BSA was used to evaluate the protein-capturing capacity of the nanoparticles [25,26,27]. As shown in Figure 3a, while the initial MNPs and MNPPEI with 2.28 mg only had a protein-loading capacity < 11%, MNPPBA with this dose had a capacity of 51% (Figure 3a). With the increase in nanoparticle masses, the three kinds of nanoparticles captured increased levels of BSA, and MNPPBA always captured the highest levels of the proteins at different masses (Figure 3b). Fluorescence microscopy was further performed to observe protein capture by the nanoparticles [28]. While both MNP and MNPPEI only adsorbed very low levels of FITC-labeled BSA (FITC-BSA), MNPPBA adsorbed drastically high levels of the FTIC-BSA, with the nanoparticles exhibiting strong green fluorescence (Figure 3c). These results indicate that MNPPBA had a strong protein-capturing capacity, which may be attributed to exposure to phenylboronic acid groups on the nanoparticle surface.

### 2.3. Efficient Enrichment of Enzymes in Rhizosphere Soil Supernatants by the Synthesized Magnetic Nanoparticles

After the higher protein-capturing capacity of MNPPBA was verified, the urease activities of the pellet and the supernatant were detected when using synthesized nanoparticles in the urease solution (100 U) [29,30]. As shown in Figure 4a, over 90% of the urease was captured by MNPPBA, about 14% captured by MNPPEI, and most of the urease was left in the supernatant of MNPs. The higher the nanoparticle dose added, the more urease captured (Figure 4b). In addition, the highest urease activity of MNPPBA pellets reached 92% when 2.28 mg of the nanoparticles were added. Similar detection results of invertase and alkaline phosphatase activities are shown in Figure 4c,d. About 93% invertase and 91% alkaline phosphatase could be captured by MNPPBA [31,32]. These results indicate that the synthesis strategy of MNPPBA was effective, and enriching trace soil enzymes by MNPPBA was feasible.

### 2.4. Sensitive Rhizosphere Enzyme Assay Based on the Synthesized Magnetic Nanoparticles

The actual soil situation is complex, meaning that the capture process of enzymes by magnetic nanoparticles may be affected [33,34]. When using *Amaranthus hypochondriacus* rhizosphere soil as the detection object, MNPPBA still showed the highest enzyme capture efficiency. Soil particles and other compositions did not inhibit the combination between nanoparticles and enzymes [35]. As shown in Figure 5, the great differences of the detection results between traditional and nanoparticle-based assay indicate that traditional assay could not reflect the soil enzymes activities accurately, especially when detecting the small amount of sample containing trace enzymes, such as plant root tissue.

The results in Figure 5 also show that nanoparticle-based assay was more sensitive and more accurate and could accurately reflect the slight changes in enzyme activities in a tiny area of the soil. It is helpful to research the response mechanism of plants and microorganisms to the changes in the soil ecological environment, as well as the role of the rhizosphere microbial system in different growth stages of plants [36].

## 3. Materials and Methods

### 3.1. Chemical Agents

Ammonium hydroxide, (3-aminopropyl)triethoxysilane, sodium hydroxide, ferrous chloride, ferric chloride, glutaraldelyde, polyethyleneimine (PEI), and 4-formylphenylboronic acid (4-FPBA) were purchased from Sigma (St. Louis, MO, USA). All reagents were used without further purification.

### 3.2. Synthesis of the Magnetic Nanoparticles

The initial magnetic nanoparticles (MNPs) were synthesized by using the co-precipitation method. Briefly, 0.4 g of ferrous chloride and 1.1 g of ferric chloride were dissolved in distilled water by magnetic stirring under 80 °C. 5 mL of ammonium hydroxide were then added into the solution. After further stirring for 1 h, the MNPs were harvested by magnetic separation by an NdFeB magnet, washed five times with distilled water, and dried with a vacuum freezing drier. The obtained MNPs were then suspended in 20 mL of ethanol. After addition of 5 mL APTES, the mixture was magnetically stirred at room temperature for 24 h. The amino group-exposed MNPs were obtained by magnetic separation and further suspended in 20 mL PEI solution (5%, *w*/*v*, prepared in distilled water). The suspension was further magnetically stirred for 4 h at room temperature. The PEI-grafted MNPs (MNPPEI) were separated by an NdFeB magnet and further suspended in 20 mL of methanol. After addition of 0.1 g of 4-FPBA, the mixture was stirred at room temperature for 12 h. The final PBA-grafted MNPs (MNPPBA) were separated by a magnet, washed twice with ethanol, and dried with a vacuum freezing drier for further characterization.

### 3.3. Characterization of the Magnetic Nanoparticles

The morphology of the nanoparticles was characterized by TEM (Tecnai G2 F-20, FEI, Hillsboro, OR, USA). Zeta potentials of the nanoparticles were detected by a dynamic light scattering (Zetasizer Nano ZS0303081003, Malvern Panalytical, London, England). FT-IR spectra of the samples were obtained by using an FT-IR analyzer (Bio-rad, Hercules, CA, USA). The X-ray diffraction spectra of the nanoparticles were characterized by an XRD analyzer (X’Pert PRO MPD, PANalytical, Almelo, The Netherlands).

### 3.4. Protein-Capturing Assays

To evaluate the capacity of the nanoparticles to capture the model protein BSA, 1 mL of BSA solution (100 μg/mL) was mixed with the three kinds of nanoparticles (i.e., MNP, MNPPEI, MNPPBA) with different doses (0.38, 0.76, 1.52, 2.28 mg/mL), respectively. After co-incubation of the mixture with gentle shaking at room temperature for 1 h, the nanoparticles were separated by a magnet. The contents of the remaining protein were detected by the Coomassie brilliant blue agent. The protein-loading capacity of the nanoparticles (%) was calculated by the adsorbed BSA divided by the mass of the nanoparticles × 100. To observe the FITC-BSA-captured nanoparticles, the nanoparticles were added into the FITC-labeled BSA (1 mg/mL). After co-incubation of 1 h, the nanoparticles were magnetically separated, washed five times with distilled water, and observed by a fluorescence microscope (BX43, Olympus, Tokyo, Japan).

### 3.5. Enzyme-Capturing Assays

To evaluate the capacity of the nanoparticles to capture the model enzymes, the solutions of the three enzymes, including urease, invertase and alkaline phosphatase, were prepared to a final concentration of 100 U/mL. The nanoparticles with different masses were then added into 1 mL of each enzyme. After 10 min of co-incubation, the nanoparticles were magnetically separated. The enzyme activity of the nanoparticles and the supernatants was detected by using assay kits of urease, invertase and alkaline phosphatase, respectively (Solarbio, Beijing, China).

### 3.6. Enrichment of Enzymes in Rhizosphere Soil Supernatants and Enzyme Assays

The *Amaranthus hypochondriacus* rhizosphere soil suspensions were prepared by washing *Amaranthus hypochondriacus* roots (1 g) with 10 mL of distilled water. 1 mL of the rhizosphere soil suspensions was then mixed with 2.28 mg of the nanoparticles. After co-incubation for 10 min, the nanoparticles were separated. The activity of the enzymes was detected by using corresponding assay kits. The enzyme activity of the soil suspensions was also detected by the traditional method, with 100 μL of the suspensions sampled for assays.

### 3.7. Statistical Analysis

Each experiment was performed with three replicates, and the values were shown with means ± SD. Differences between groups were compared by a Student’s *t*-test (*p* < 0.05). All statistical tests were performed using the SPSS software package (Version 20, IBM, Armonk, NY, USA).

## 4. Conclusions

This study showed the synthetic method of magnetic nanoparticles with high soil enzyme-capturing capacity and verified the effect of phenylboronic acid groups on the nanoparticle surface. More than 90% of the main soil enzymes captured by MNPPBA showed its strong effectiveness in the enrichment of trace enzymes. The nanoparticle-based assay of soil enzyme holds significant advantages in sensitivity and accuracy and also improves the enrichment efficiency of trace enzymes and reduces the pretreatment workload and detection costs, showing a promising method for soil enzyme activity detection.

## Figures and Tables

**Figure 1 molecules-27-06883-f001:**
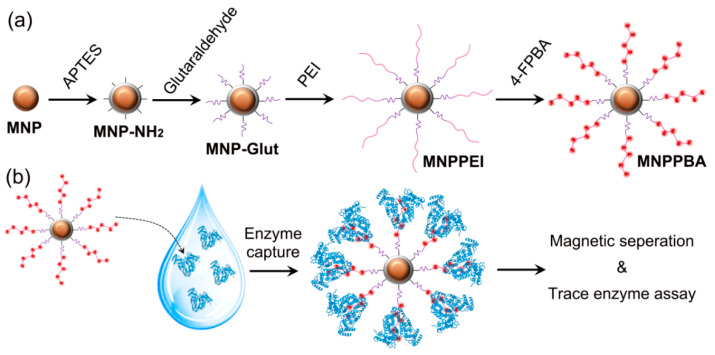
Schematic illustration of MNPPBA synthesis (**a**) and its application in trace enzyme assay (**b**).

**Figure 2 molecules-27-06883-f002:**
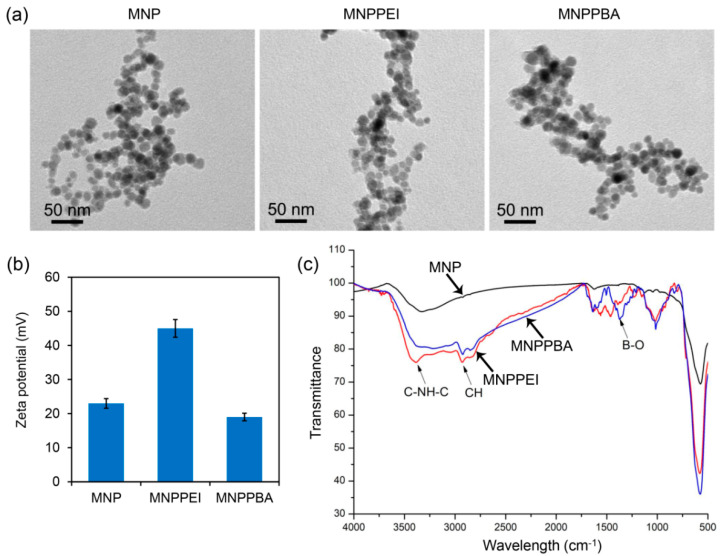
Characterization of MNP, MNPPEI and MNPPBA. (**a**) TEM images of the synthesized nanoparticles. (**b**) Zeta potentials of the nanoparticles. (**c**) FT-IR spectra of the nanoparticles.

**Figure 3 molecules-27-06883-f003:**
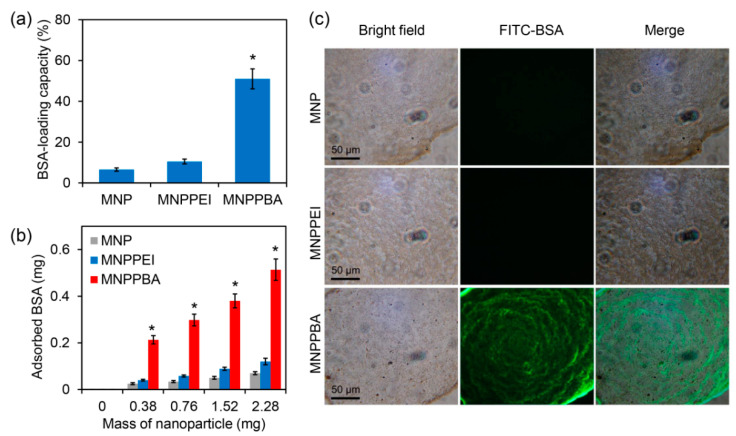
BSA capture by the synthesized nanoparticles. (**a**) BSA loading capacity of the nanoparticles (2.28 mg/mL) with the BSA solution (1 mg/mL) after 5 min of co-incubation. (**b**) BSA adsorption by the nanoparticles with different doses. (**c**) Fluorescence microscopy images of the nanoparticles after 5 min of co-incubation with the solution of FITC-labeled BSA. The asterisks (*) indicate a significant difference between MNPPBA and the other two groups (*p* < 0.05).

**Figure 4 molecules-27-06883-f004:**
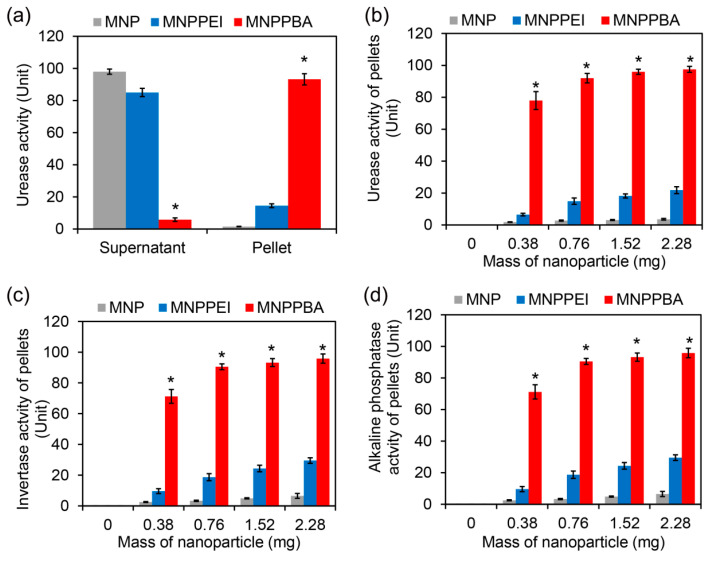
Enzyme capture and magnetic enrichment by the synthesized nanoparticles. (**a**) Urease activity of the supernatant and the nanoparticle pellets. 2.28 mg of the nanoparticles were added into 1 mL of the urease solution (100 U). After 5 min of co-incubation, the nanoparticles were separated by an NdFeB magnet. The nanoparticle pellets and the supernatants were used for detection of urease activity by a urease assay kit. (**b**) Urease activity of the nanoparticle pellets at different nanoparticle doses. (**c**) Invertase activity of the nanoparticle pellets at different nanoparticle doses. (**d**) Alkaline phosphatase activity of nanoparticle pellets at different nanoparticle doses. The asterisks (*) indicate a significant difference between MNPPBA and the other two groups (*p* < 0.05).

**Figure 5 molecules-27-06883-f005:**
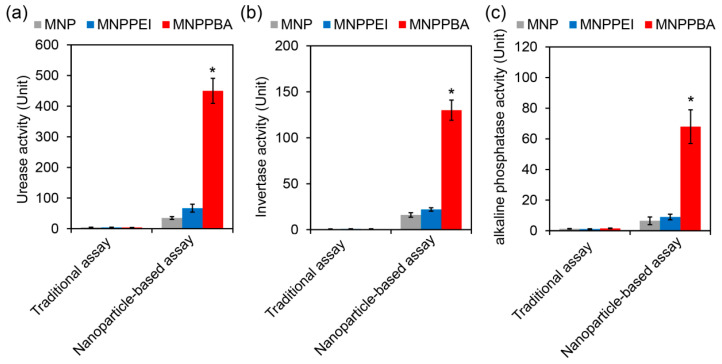
Enzyme activity of *Amaranthus hypochondriacus* rhizosphere soil supernatants detected by the traditional assay method and by the nanoparticle-based assay. (**a**) Urease activity. (**b**) Invertase activity. (**c**) Alkaline phosphatase activity. The asterisks (*) indicate a significant difference between MNPPBA and the other two groups (*p* < 0.05).

## Data Availability

Not applicable.

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
