# Peer review of "Synthesis of Phenylboronic Acid-Functionalized Magnetic Nanoparticles for Sensitive Soil Enzyme Assays"

_molecules, 2022, doi:10.3390/molecules27206883_

Round 1
Reviewer 1 Report
The paper has been written very well by the authors and I don't feel there is any change required in the paper.
Author Response
Thanks for your affirmation of this paper.
Reviewer 2 Report
English editing is required. Discussion could be improved.
Author Response
Thanks for your comments and suggestions. English of the paper and discussion were improved.
Reviewer 3 Report
Soil enzyme is an important active component in soil, so it is very important to establish a rapid and accurate method to detect soil enzyme. This research is of great significance, but it needs to be revised before publication.
1, This paper does not fully review the previous research progress. What are the shortcomings of the current soil enzyme testing methods? What efforts have the predecessors made for these shortcomings? What is the scientific problem of this paper?
2, The results in Fig. 5 show that the enzyme activity measured by the magnetic nanoparticles method is much higher than that measured by the traditional test method. How can the author ensure the accuracy of which method? What is your verification method?
3, The conclusion should be further enriched. The author should also give a prospect of the application prospect of this method. What is the biggest limitation of the promotion and application of this method? Such as stability, accuracy or cost?
4, The format of the article should be carefully checked. For example, some references use the full name of the magazine, but some are abbreviated.
Author Response
Thanks for your comments and suggestions.Please see the attachment.
